# Reduced Use of Nitrites and Phosphates in Dry-Fermented Sausages

**Martin Škrlep [1],\*** , **Manja Ozmec [2]** and **Marjeta Čandek-Potokar [1,2]**

1   Agricultural Institute of Slovenia, Hacquetova ulica 17, 1000 Ljubljana, Slovenia; meta.candek-potokar@kis.si
2   Faculty of Agriculture and Life Sciences, University of Maribor, Pivola 10, 2311 Hoče-Slivnica, Slovenia; manja.ozmec@gmail.com
\*   Correspondence: martin.skrlep@kis.si

**Abstract:** Given consumer demand for foods with fewer artificial additives, the objective of this study was to investigate the effects of reduced use of nitrites and phosphates on dry-fermented sausage quality. Four sausage formulations were prepared: (1) control (using standard procedure with 0.2% phosphates and 110 mg/kg sodium nitrite) and formulations with (2) 50% less sodium nitrite, (3) 50% less sodium nitrite and sodium ascorbate (225 mg/kg), and (4) with standard nitrite but no phosphates. Weight loss and pH evolution were monitored during processing. The color, physicochemical (including oxidation), rheological, and sensory properties were evaluated on the finished product, as well as mold growth and microbiological status. Compared to control, nitrite reduction was associated with increased surface mold growth, reduced (3.0–4.4%) processing loss, and slightly higher oxidation (1.7 μg/kg more malondialdehyde) but without affecting instrumental color. The simultaneous addition of ascorbate reduced oxidation and improved color stability. The formulation without the phosphates resulted in increased oxidation (3.4 μg/kg more malondialdehyde) and changes in the instrumental color. The observed changes were relatively unimportant, as neither of the tested formulations influenced sensory traits or compromised microbial safety, implying that they can be used in production without any harm or even with some benefits.

**Keywords:** dry sausages; additives; nitrites; phosphates; ascorbates

## 1. Introduction

The properties of dry-fermented sausages depend on the quality of the raw material, the added ingredients, and the processing conditions. During processing, intense physical (especially dehydration), biochemical (proteolysis, lipolysis, oxidation), and microbiological changes (growth and metabolic activity of certain types of microorganisms) take place which affect the development of sensory characteristics of the final product [1,2] and ensure the stabilization of the product during prolonged storage [3]. The above processes are strongly influenced by various additives which are indispensable in the modern food industry, as they have many beneficial effects (e.g., ensuring microbiological stability, antioxidant activity, appropriate organoleptic properties, etc. [4]), although they may also have some adverse effects [5].

Nitrites and nitrates are among the most important additives used in meat processing. Nitrates have no direct effect but are converted by the microbial enzyme nitrate reductase to nitrite, which is highly reactive with proteins and other chemical compounds. Its main function is antimicrobial, acting against numerous aerobic or anaerobic microorganisms, including *Clostridium botulinum* and pathogens such as *Listeria monocytogenes*, *Staphylococcus aureus*, or *Clostridium perfringens* [6–8]. In addition, nitrite slows down oxidative processes during prolonged storage and/or thermal treatment (thus preventing the development of undesirable aroma and taste perceived as rancid [9]). It also provides stable color of cured meat due to the reaction of $NO^+$, produced by further reduction of

nitrite and muscle pigment myoglobin, and ensures unique flavor of the cured meat [7]. On the other hand, several negative (toxic) effects of nitrite also occur. Part of the nitrite can be reoxidized back to nitrate, which, in the presence of organic matter and favored by low pH and higher temperatures, reacts with amines to form nitrosamines, which are potent carcinogenic compounds, while excessive levels of nitrite can also lead to the degradation of erythrocytes and vitamin A [10,11]. The occurrence of nitrosamines can be reduced by the addition of ascorbic acid. Due to its role as a chemical reducing agent, ascorbic acid or its salts accelerate nitrite reduction and the formation of the stable meat pigment nitrosomyoglobin, while its general role as an antioxidant prevents oxidation and discoloration during prolonged storage of meat products [12–14] and increases the antibacterial activity of nitrite [6]. Apart from very high doses, which are almost impossible to achieve in practice, no toxic effects of ascorbates are reported [15], and they could serve as a more natural (i.e., ascorbic acid, also called vitamin C, occurs naturally in foods) substitute antioxidant for nitrites. However, the antioxidant effect of ascorbate has not been fully elucidated, either in terms of the dose or in combination with nitrite [16,17]

Another important group of additives are phosphates, whose main benefit is the improved water binding capacity of meat, increasing the technological value of processed meat [4]. In addition, phosphates act as emulsifiers, are involved in color development, and, to a lesser extent, exert antimicrobial and antioxidant effects [4]. On the other hand, excessive addition of phosphates may be associated with the formation of a bitter, soapy, or metallic aftertaste, a very firm texture, and possible indirect negative effects on human health, i.e., an imbalance of calcium and phosphorus. Although phosphates are mainly used in thermally processed meat products, they are also added to dry sausages to ensure good adhesion and texture of the meat batter [18]. According to Walz et al. [19] the addition of phosphates can also prevent the formation of white surface efflorescence, which is perceived negatively by consumers. Nevertheless, the use of phosphates in dry sausages seems to be somewhat controversial, as their water binding potential and alkaline pH may interfere with the drying and acidification processes, which are important for the normal processing of dry sausage, necessitating further evaluation of their usefulness as additives in this product category.

To counter the negative effects of chemical additives, which in turn are reflected in negative consumer acceptance, many natural compounds have been proposed [5], including various organic acids, spices, and fruit or vegetable extracts [5,20]. However, many disadvantages, such as higher cost and especially lower efficacy [21,22], significantly limit their use. Therefore, one of the current trends aims at reducing the amount of additives used or replacing them with less toxic ones (e.g., ascorbates) [23,24]. However, their reduction is also associated with quality changes or reduced microbiological safety, as shown, for example, in the case of nitrites [25–27]. Any change in already established meat processing methods requires careful consideration of the impact on the different quality levels [28,29]. While there are numerous studies addressing the possibility of additive alterations, most of them focus on a relatively narrow range of traits, such as microbiological profile [25–27], and chemical changes, such as oxidation or lipolysis [17,30], or do not include aspects such as sensory characteristics [31]. In line with the tendency to reduce additives in meat products, the aim of the present study was to investigate how the reduction of nitrites and phosphates and the addition of ascorbate (as a replacement antioxidant) affect the processing dynamics and final quality of dry-fermented sausages. The study included the evaluation of four different formulations with different levels of nitrites, phosphates, and ascorbate and considered a variety of studied characteristics during processing and on the final product in terms of physicochemical, sensory, and textural properties as well as the evaluation of the microbiological safety of the tested products.

## 2. Materials and Methods

### 2.1. Dry-Fermented Sausages Formulations and Processing

The raw material (meat and fat) used to make the sausages originated from commercially fed and reared pigs of modern crossbreeds. The chilled hind leg muscle meat and backfat (in a 4:1 weight ratio) were minced with a 10 mm mincing plate. Four different formulations were formed, each made in two consecutive batches (i.e., technical replicates), consisting of 12–15 sausages for each of the batches, for a total of 24–30 for each formulation. For the control formulation (CON), the meat batter was mixed with 2.2% curing salt (TKI Hrastnik, d.d., Hrastnik, Slovenia; consisting of sodium chloride with 0.5% sodium nitrite, resulting in a concentration of 110 mg sodium nitrate per kg of meat batter) and 0.2% phosphate mixture according to the manufacturer's instructions (Sofos 4×, TKI Hrastnik, d.d., Hrastnik, Slovenia; containing 58% di-, tri-, and polyphosphates expressed as $P_2O_5$ equivalents). The first experimental formulation (50NI) contained 0.2% phosphate mixture, with the addition of sodium nitrite reduced by 50% (by adding 1.1% pure sodium chloride and 1.1% curing salt). The second experimental formulation (50NA) was prepared in a similar manner with a 0.2% phosphate mixture and a 50% nitrite reduction but with the addition of 225 mg/kg sodium ascorbate (TKI Hrastnik, d.d., Hrastnik, Slovenia). The third experimental formulation (NP) consisted of the addition of 2.2% curing salt (as in the control formulation) but without the addition of the phosphate mixture. The ratios of additives in sausage formulations were by weight. Equal amounts of dried garlic (0.5%), cumin (0.2%), and pepper (0.3%) were used in all formulations. After the batter was thoroughly mixed, it was stuffed into natural casings (3 cm in diameter). The sausages were hung on a rack and dried at a relative humidity of 65–68% and a temperature of 8–9 °C for 24 days until an average weight loss of 45%. During processing, weight loss was monitored weekly. The pH was monitored by measuring it daily during the first week and weekly thereafter using the MP120 pH meter (Mettler-Toledo GmbH, Schwarzenbach, Switzerland). The number of samples used for specific analysis is shown in Figure 1.

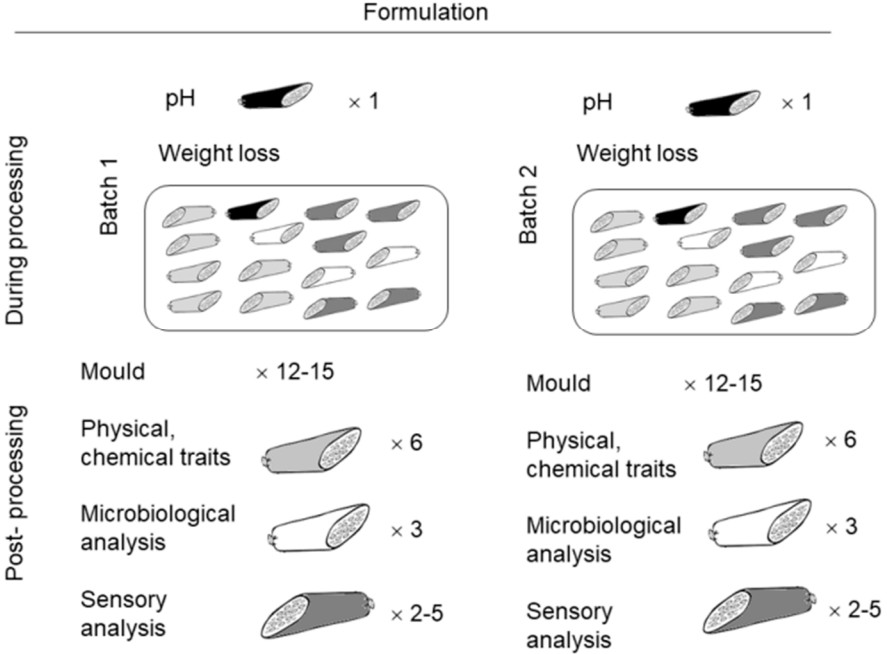

**Figure 1.** Number of sausages used for specific analyses.

### 2.2. Microbiological Parameters

At the end of processing, the extent of mold growth on the surface was evaluated, using a rating scale of 1–5, with a score of 1 representing no mold growth and a score of 5 representing complete mold coverage. Scoring was performed by two evaluators, and the average of

both scores was used for data analysis. The sausages were then packed in vacuum bags and frozen at −20 °C until further analysis. Microbiological analyses were performed by an accredited laboratory (National Laboratory for Health, Environment, and Food, Murska Sobota Department, Murska Sobota, Slovenia) according to nationally recognized guidelines for microbiological food safety [32] and included bacteriological culture tests for the detection of *Salmonella* spp. [33], *Clostridium perfringens* [34], *Yersinia enterocolitica* [35], *Listeria monocytogenes* [36], *Staphylococcus aureus* and other coagulase-positive cocci [37], and *Enterobacteriae* [38].

### 2.3. Measurements of pH and Color

For pH measurement, 5 g of homogenized sausage sample was mixed in 20 mL of distilled water [39]. Objective color parameters (in the CIELAB color system, where L* is defined as lightness and a* and b* are defined as redness and yellowness, respectively) were measured using the Minolta Chroma Meter CR300 (Minolta Co., Ltd., Osaka, Japan) with an 8 mm aperture and a D65 illuminant [40]. The cured sausages were cut into 1 cm thick slices, and the color was measured on the freshly cut cross-section. To simulate the effect of storage conditions on color, the sliced samples were vacuum packed and stored in a refrigerator at 4 °C and then exposed to air for 1 h. In both cases (i.e., freshly sliced and after storage), 12 to 14 color measurements (corresponding to each slice) were taken for each sausage and averaged for data analysis.

### 2.4. Instrumental Texture Measurement

For the instrumental texture measurements, 15 mm thick cylinders with a diameter of 25 mm were cut from the middle part of the sausages. Texture profile analysis (TPA) and stress relaxation tests (SR) were performed in triplicate per sausage, as described by Pugliese et al. [41], using the TA Plus texture analyzer (Ametek Lloyd Instruments, Ltd., Bognor Regis, West Sussex, UK). Briefly, for the TPA the samples were compressed twice to 50% of their original height; the time–force curves were recorded and used to calculate the individual rheological parameters (i.e., hardness, cohesiveness, gumminess, chewiness, springiness, and adhesiveness). In the case of the SR test, the force decay coefficient was calculated from the measurement of the initial force and the force recorded after the sample had been compressed by 25% after 90 s of compression.

### 2.5. Determination of Chemical Composition

Prior to chemical composition determination, samples were manually cut into small pieces, frozen in liquid nitrogen, and homogenized into fine powder using an IKA M20 mill (IKA-Werke GmbH & Co., Staufen, Germany). Chemical composition (moisture, lipids, proteins, salt) was evaluated by near-infrared spectroscopy (NIR Systems 6500 Monochromator, FOSS NIR System, Silver Spring, MD, USA) as described by Prevolnik et al. [42], using internal calibrations developed at the Agricultural Institute of Slovenia. Water activity ($a_w$) was measured in 10 g of the sample using Aqualab 4TE (Meter Group Inc., Pullman, WA, USA). The degree of lipid oxidation was measured by thiobarbituric acid reactive substances (TBARS) analysis according to the method described by Lynch and Frei [43]. For this analysis, 1 g of minced samples was homogenized for 20 s in 10 mL of 0.15 M KCl solution containing 0.1 mM butylated hydroxytoluene using the UltraTurrax T25 disperser (IKA-Werke GmbH & Co., Staufen, Germany). The resulting homogenate was centrifuged (Hereaus Megafuge 8R, Termo Fischer Scientific, Waltham, MA, USA) at 3600 rounds per minute for 12 min. The resulting supernatant (0.5 mL) was then incubated at 100 °C for 10 min with 0.25 mL of 2.8% (*w/v*) trichloroacetic acid and 0.25 mL of 1% (*w/v*) 2-thiobarbituric acid in 50 mM NaOH using a thermostatic heating block (TBGS-b, Marijan Krokter s.p., Ljubljana, Slovenia). The sample was cooled to room temperature, and 2 mL of n-butanol was added to allow the extraction of the resulting pink chromogen. After stirring in a vortex mixer (Vibromix 10, Tehtnica Železniki d.o.o., Železniki, Slovenia) for 10 s and centrifugation (10 min at 1500 rounds per minute) the absorbance of

the upper (n-butanol) phase was measured at 535 nm using the Infinite 200 Pro Reader (Tecan Group Ltd., Männedorf, Switzerland). TBARS concentration was calculated using 1,1,3,3-tetraethoxypropane as a standard and expressed as malondialdehyde concentration (in μg per kg sample).

### 2.6. Sensory Analysis

Sensory analysis was performed using a quantitative descriptive analysis [44] with six trained panelists, four female and two male, 41–69 years old, all nonsmokers. Prior to testing, the panel was trained on a variety of commercially available dry-fermented sausages characterized by a wide range of maturity, dryness, tastes (including saltiness), texture, and color. Eleven different sensory descriptors were defined to describe appearance (homogeneity, intensity, and brightness of color), intensity of typical dry-cured aroma, taste descriptors of saltiness, off-tastes, and rancidity and texture descriptors of softness, crumbliness, juiciness, and pastiness. Each sensory descriptor was rated on a 9 cm unstructured scale anchored at the two extremes ("not perceived" and "very intense" on the left and right sides, respectively). Panelists abstained from food and drink (except water) for two hours before analysis and were not given any information about the nature of the study. Panelists received samples in two rounds consisting of two sausage slices (3 mm thick) on a white plate coded with three digits. Between tastings of each sample, water and bread were offered for sensory neutralization.

### 2.7. Statistical Analysis

Statistical analysis was performed using SAS statistical software (SAS Institute Inc., Cary, NC, USA). In the case of pH and processing loss, statistical analysis was not possible, because the data were collected based on the formulation. For other parameters, including color, texture, and physicochemical properties, the data were analyzed by the GLM procedure, using the model with the formulation and technical repetition as fixed effects. Interaction was also tested but excluded due to statistical insignificance. For sensory properties, a repeated measures analysis (within sample, with panelist as repetition) was performed using the MIXED procedure, with the model including formulation as a fixed effect and panelists as a random effect. When formulation effect was significant ($p < 0.05$), least squares means were compared using Dunnett test (comparison with control).

## 3. Results and Discussion

### 3.1. Changes during Processing (pH, Weight Loss), Sausage Surface, and Microbiological Profile

The measurement of pH during processing showed a steady increase until the second week, after which a slight pH decrease was observed in the last week (Figure 2). The evolution of pH followed the same pattern for all formulations, although the sausages of the NP group showed slightly lower values throughout the processing. This result is likely due to the absence of phosphates, which are known to have an alkaline pH [45]. Regardless of the formulation, all sausages showed a slight increase in pH, which is an expected change in dry-fermented sausages without added sugar. This can be attributed to the fact that during a slow-onset proteolytic process, non-protein nitrogen compounds such as polypeptides, amines, and ammonia are formed by the action of endogenous proteolytic enzymes as well as various microorganisms such as bacteria, molds, and yeasts [1].

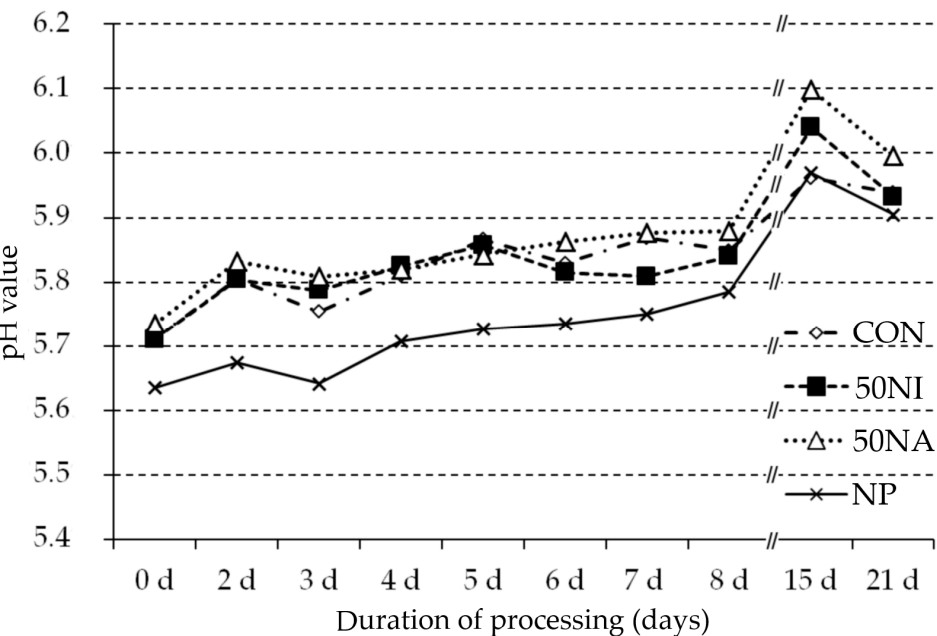

**Figure 2.** Changes in pH during the sausage processing according to formulation (CON = control with 110 mg/kg sodium nitrite and 0.2% phosphates; 50NI = 50% less sodium nitrite than in control; 50NA = 50% less sodium nitrite than in control with added 225 mg/kg sodium ascorbate; NP = no phosphates and 110 mg/kg sodium nitrite).

With regard to processing loss (Figure 3), the lowest values were observed in both formulations with reduced nitrite addition (3.0 to 4.0% and 4.0 to 4.4% lower loss in 50NI and 50NA, respectively), while the processing loss in the NP group was similar to CON. Reduced nitrite addition (i.e., in 50NI and 50NA) was also associated with higher ($p < 0.001$) surface mold growth compared to both treatments (CON, NP) with standard nitrite content (Figure 4, Table 1). Microbiological testing of the sausages yielded negative results for the presence of harmful bacteria in all formulations (i.e., no presence of *Salmonella* spp., Yersinia enterocolitica, and Listeria monocytogenes in 25 g, less than 100 CFU (colony forming units) per g of Staphylococcus aureus and other coagulase-positive cocci, and less than 10 CFU per g of Clostridium perfringens and Enterobacteriae). Nevertheless, microorganisms probably played an important role in sausage processing, namely with relation to moisture loss. We observed increased growth of surface microorganisms in both nitrite-reduced treatments (50NI, 50NA), possibly related to the lower concentrations of nitrite, which is known for its antimicrobial activity [7–9]. The presence of molds and yeasts on the surface of meat products has been shown to reduce water loss via evaporation. The study by Corral et al. [46] showed that microbial inoculation with the yeast Debaryomyces hansenii successfully reduced excessive dehydration and effectively prevented associated textural defects. Similarly, Müller et al. [47] demonstrated that sausages with mold on the surface lost less water and were up to 30% softer than sausages without mold. Contrary to our expectations, the presence of phosphates did not have such an effect on weight loss, implying that the addition of phosphates to the batter did not improve moisture retention during sausage processing. This finding confirms the results of Szmanko et al. [48], who compared different sausage formulations of "Krakowska" dry sausage and also found no effect of phosphate addition on moisture content. The reason for the lack of effect of phosphates could be the relatively high pH and salt content. According to Xiong [4], phosphates lose their water binding ability at pH values above 5.5–6.0 and salt level above 2%, both conditions that were present in this study. Even though a 50% nitrite reduction was sufficient to increase the growth of molds on the surface of the sausages, this reduction still did not allow the development of potentially harmful or toxic bacteria. According to EFSA opinion [23], microbial safety of long-matured meat products with high pH and low

salt content can only be achieved after addition of up to 150 mg nitrite per kg of meat. At 60 mg/kg, both the 50NI and 50NA formulations were below this limit, but factors such as rapid desiccation and salt concentration lower the $a_w$ value to the point of safety. According to Toldrá and Flores [1], the $a_w$ value of 0.89 ensures microbial stability. At the same time, the addition of ascorbate further improves antimicrobial activity by sequestering metal ions [49].

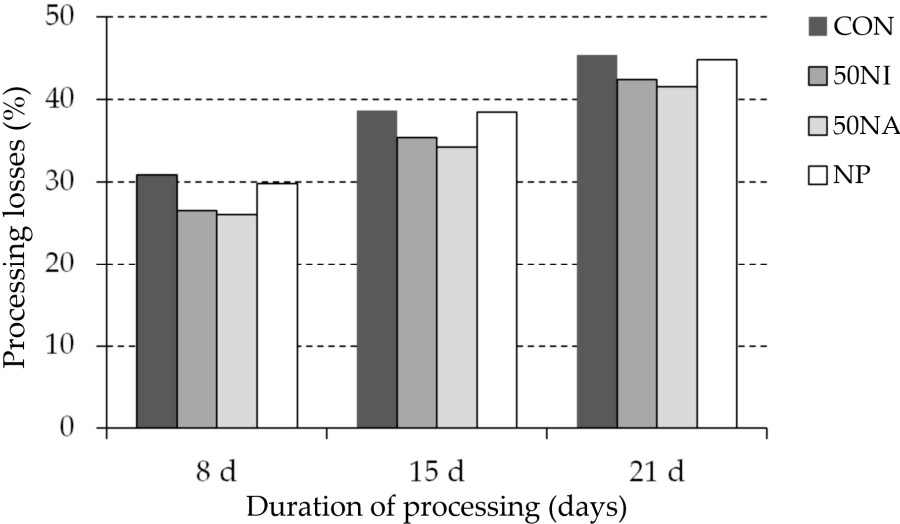

**Figure 3.** Processing loss (in %) according to formulation (CON = control with 110 mg/kg sodium nitrite and 0.2% phosphates; 50NI = 50% less sodium nitrite than in control; 50NA = 50% less sodium nitrite than in control with added 225 mg/kg sodium ascorbate; NP = no phosphates and 110 mg/kg sodium nitrite).

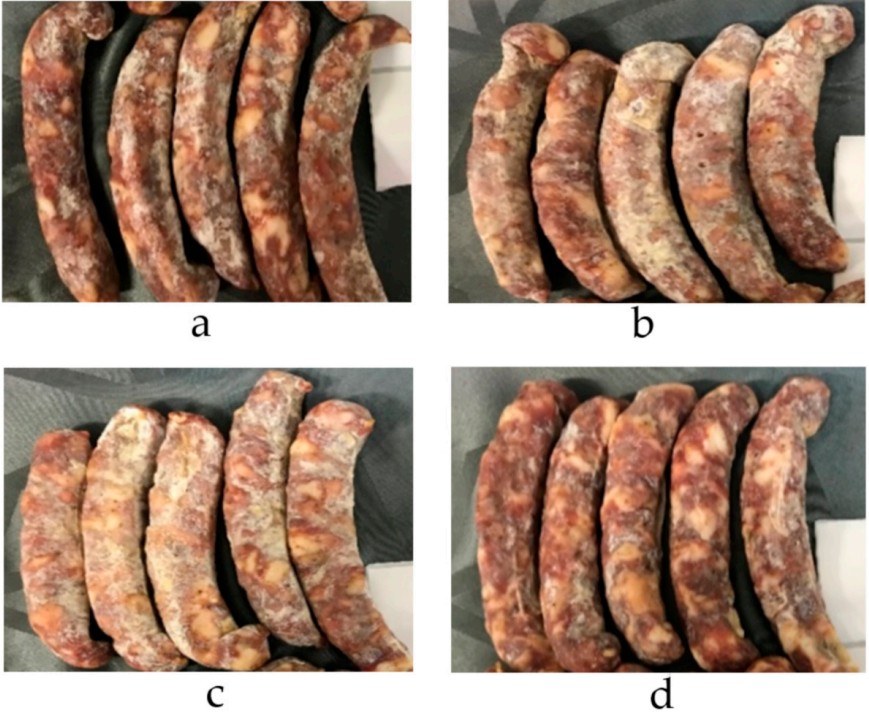

**Figure 4.** Surface mold covering of sausages ((**a**) = control with 110 mg/kg sodium nitrite and 0.2% phosphates; (**b**) = 50% less sodium nitrite than in control; (**c**) = 50% less sodium nitrite than in control with added 225 mg/kg sodium ascorbate; (**d**) = no phosphates and 110 mg/kg sodium nitrite).

**Table 1.** Evaluation of mold covering of the sausage surface (visual assessment on a scale 1–5).

| | CON | 50NI-CON [a] | 50NA-CON [a] | NP-CON [a] | RMSE | Effect of F |
|---|---|---|---|---|---|---|
| Mold covering | 1.8 | 2.5 *** | 2.7 *** | −0.3 NS | 0.67 | *** |

[a] The results are presented as differences (least squares means) between the control group and the tested formulations. CON = control with 110 mg/kg sodium nitrite and 0.2% phosphates; 50NI = 50% less sodium nitrite than in control; 50NA = 50% less sodium nitrite than in control with added 225 mg/kg sodium ascorbate; NP = no phosphates and 110 mg/kg sodium nitrite; RMSE = root mean square error of the model. F = formulation; significance: NS = $p > 0.10$; *** = $p < 0.001$.

## 3.2. Instrumental Color Parameters

As shown in Table 2, reducing nitrite had no effect ($p > 0.10$) on the objective color parameters measured on the freshly cut sausage surface (i.e., 50NI and 50NA did not differ from CON), while the absence of phosphates resulted in significantly darker ($p < 0.05$), less red, and less yellow color ($p < 0.01$), namely, lower L*, a*, and b* in the NP group compared to the CON group. In the color measurements after storage and air exposure, the 50NI sausages were not significantly different ($p > 0.10$) from CON, while the addition of ascorbate resulted in redder ($p < 0.05$) and yellower ($p < 0.001$) color in 50NA compared to CON. The sausages in the NP group also had significantly less red ($p < 0.01$) and tended to have ($p < 0.10$) less yellow color compared to CON. This result is consistent with the fact that a relatively low concentration of nitrites is required for reaction with myoglobin, which stabilizes the color. It has been found that as little as 25 ppm of nitrites is sufficient for color stabilization, which is less than half the added concentration in 50NI and 50NA sausages [50]. On the other hand, the much more pronounced effect of omitting phosphates on color may be related to the higher degree of oxidation in NP sausages (namely TBARS), which probably caused oxidation of meat pigments and consequent discoloration [51]. In the case of color measurements after prolonged storage and air exposure, similar conclusions can be drawn, while the effect of ascorbate addition (higher a* and b* color parameters) can be explained by the known relationship between ascorbate and meat color stability [12,13].

**Table 2.** Instrumental color parameters of dry-fermented sausages.

| | CON | 50NI-CON [a] | 50NA-CON [a] | NP-CON [a] | RMSE | Effect of F |
|---|---|---|---|---|---|---|
| Fresh cut | | | | | | |
| L* | 43.2 | 1.5 NS | −2.1 NS | −3.6 * | 3.27 | ** |
| a* | 8.1 | −0.7 NS | 0.7 NS | −2.0 ** | 1.60 | ** |
| b* | 5.9 | 0.2 NS | −0.1 NS | −1.2 ** | 0.72 | *** |
| After storage and air exposure | | | | | | |
| L* | 42.8 | −0.3 NS | −1.0 NS | −1.9 NS | 2.38 | NS |
| a* | 8.5 | −0.4 NS | 1.1 * | −1.4 ** | 0.97 | *** |
| b* | 5.8 | 0.1 NS | 0.7 *** | −0.4 ‡ | 0.44 | *** |

[a] The results are presented as differences (least squares means) between the control group and the tested formulations. CON = control with 110 mg/kg sodium nitrite and 0.2% phosphates; 50NI = 50% less sodium nitrite than in control; 50NA = 50% less sodium nitrite than in control with added 225 mg/kg sodium ascorbate; NP = no phosphates and 110 mg/kg sodium nitrite; RMSE = root mean square error of the model. F = formulation; significance: NS = $p > 0.10$; ‡ = $p < 0.10$; * = $p < 0.05$, ** = $p < 0.01$; *** = $p < 0.001$.

## 3.3. Sausage Physical Chemical Traits

Regarding chemical composition (Table 3), no significant differences ($p > 0.10$) were found between CON and the other sausage formulations in terms of lipid and protein content and proteolysis index. Sausages from 50NA tended to have ($p < 0.10$) higher moisture content and $a_w$ value ($p < 0.01$) compared to CON. NP sausages exhibited lower final pH ($p < 0.001$) and higher salt concentration CON. Consistent with the tendency toward lower moisture content, 50NA sausages had lower $a_w$ value ($p < 0.01$) than CON. Compared to CON, the degree of oxidation (TBARS) was higher in 50NI ($p < 0.05$) and in NP ($p < 0.001$) sausages. The differences in chemical composition correspond to the differences in processing loss. Despite similar processing losses in NP and CON sausages, the resulting

moisture content was slightly (but only numerically) higher in NP sausages, which could explain the observed higher NaCl content in NP than CON sausages. The differences in oxidation are also interesting, as they confirm the benefits of additives, i.e., the antioxidant properties of nitrites, ascorbates, and phosphates [50]. The decrease in nitrites in our study was proportional to the increase in oxidation, i.e., a 50% reduction was associated with 48% higher TBARS levels in 50NI than CON sausages. On the other hand, this deficiency was compensated by the addition of ascorbates to the nitrite-reduced formulation. Although the literature indicates a rather weak antioxidant potential of phosphates [4,45], it proved to be relatively important in our study. The absence of phosphates in NP sausages (with the same nitrite content as CON) resulted in more than 50% higher TBARS. The reasons for this may also be indirect. Oxidation is accelerated by lower pH [52,53], which was observed in this group (Figure 1) due to the absence of alkaline phosphates. A slightly higher concentration of salt, which acts as a pro-oxidant [54], was also observed in this group and may have contributed to higher oxidation. Overall, it should be noted that the TBARS levels observed were all well below the sensory detection limit (0.5 mg/kg) [55], which was accordingly not detected in the sensory test.

**Table 3.** Physical chemical parameters of dry-fermented sausages.

| | CON | 50NI-CON [a] | 50NA-CON [a] | NP-CON [a] | RMSE | Effect of F |
|---|---|---|---|---|---|---|
| Lipids, g/kg | 310.2 | −5.1 NS | −3.6 NS | 0.8 NS | 25.56 | NS |
| Moisture, g/kg | 309.5 | 9.2 NS | 12.6 ‡ | −9.9 NS | 13.62 | * |
| Proteins, g/kg | 319.2 | −2.3 NS | −11.1 NS | 10.8 NS | 15.88 | * |
| PI, % | 9.8 | −0.2 NS | −3.5 NS | −0.1 NS | 0.45 | NS |
| Salt, g/kg | 59.6 | 0.2 NS | −0.6 NS | 2.8 * | 2.33 | ** |
| Final pH | 6.24 | −0.01 NS | −0.00 NS | −0.20 *** | 0.06 | *** |
| $a_w$ | 0.86 | 0.02 NS | 0.03 ** | 0.01 NS | 0.01 | ** |
| TBARS, µg MDA/kg | 3.3 | 1.7 * | −1.0 NS | 3.4 *** | 1.19 | *** |

[a] The results are presented as differences (least squares means) between the control group and the tested formulations. CON = control with 110 mg/kg sodium nitrite and 0.2% phosphates; 50NI = 50% less sodium nitrite than in control; 50NA = 50% less sodium nitrite than in control with added 225 mg/kg sodium ascorbate; NP = no phosphates and 110 mg/kg sodium nitrite; RMSE = root mean square error of the model; F: formulation; PI = index of proteolysis; TBARS = thiobarbituric acid reactive substances; MDA = malondialdehyde. Significance: NS = $p > 0.10$; ‡ = $p < 0.10$; * = $p < 0.05$; ** = $p < 0.01$; *** = $p < 0.001$.

### 3.4. Sausage Rheological and Sensory Traits

Measurements of rheological properties (Table 4) showed no differences ($p > 0.10$) between formulations in the case of the SR test (i.e., the Y90 parameter), while TPA showed some, albeit relatively limited, effects. Nitrite reduction alone had no effect ($p > 0.05$) on any of the measured texture characteristics, but when accompanied by ascorbate addition, the sausages from this treatment (i.e., 50NA) tended to be more cohesive ($p < 0.10$) and chewy than CON sausages. The absence of phosphates also did not result in a significant difference; only cohesiveness tended to be lower ($p < 0.10$) in NP than in CON. Sensory analysis also revealed no additive-related difference in the sensory descriptors studied (Table 5). Only a negligible difference was found in color brightness, which tended to be higher in NP ($p < 0.10$) than in CON. In general, the effects were negligible. These results cannot be directly related to other findings, such as chemical characteristics. It is possible that the differences were too small to be noticed. Although no effects of nitrites on the activity of proteolytic enzymes have been reported in the literature, the addition of ascorbates may inhibit enzymes such as cathepsin H, m-calpain, and leucine aminopeptidase in meat during maturation [56,57].

**Table 4.** Instrumental texture parameters of dry-fermented sausages.

| | CON | 50NI-CON [a] | 50NA-CON [a] | NP-CON [a] | RMSE | Effect of F |
|---|---|---|---|---|---|---|
| Hardness, N | 186.4 | 9.0 [NS] | 5.4 [NS] | −6.5 [NS] | 28.33 | NS |
| Cohesiveness | 0.39 | 0.03 [NS] | 0.04 [‡] | −0.04 [‡] | 0.043 | *** |
| Gumminess, N | 73.1 | 8.1 [NS] | 10.3 [NS] | −9.1 [NS] | 13.71 | ** |
| Springiness, mm | 3.6 | −0.0 [NS] | 0.2 [NS] | 0.1 [NS] | 0.29 | NS |
| Chewiness, N | 258.4 | 27.5 [NS] | 44.6 [‡] | −31.5 [NS] | 47.76 | ** |
| Adhesiveness, N * mm | −2.2 | −0.2 [NS] | −0.2 [NS] | −0.2 [NS] | 0.61 | NS |
| Y90 | 0.63 | 0.01 [NS] | 0.00 [NS] | 0.01 [NS] | 0.012 | NS |

[a] The results are presented as differences (least squares means) between the control group and the tested formulations. CON = control with 110 mg/kg sodium nitrite and 0.2% phosphates; 50NI = 50% less sodium nitrite than in control; 50NA = 50% less sodium nitrite than in control with added 225 mg/kg sodium ascorbate; NP = no phosphates and 110 mg/kg sodium nitrite; RMSE = root mean square error of the model; F: formulation; Y90 = force decay coefficient. Significance: NS = $p > 0.10$, ‡ = $p < 0.10$; * = $p < 0.05$; ** = $p < 0.01$, *** = $p < 0.001$.

**Table 5.** Sensory traits of dry-fermented sausages.

| | CON | 50NI-CON [a] | 50NA-CON [a] | NP-CON [a] | RMSE | Effect of F |
|---|---|---|---|---|---|---|
| Color intensity | 6.36 | 0.41 [NS] | 0.53 [NS] | 0.73 [NS] | 1.31 | NS |
| Color brightness | 7.01 | −0.31 [NS] | 0.27 [NS] | −0.53 [‡] | 1.16 | * |
| Color uniformity | 7.69 | 0.14 [NS] | −0.01 [NS] | −0.42 [NS] | 0.95 | NS |
| Matured aroma | 7.24 | 0.46 [NS] | 0.01 [NS] | 0.04 [NS] | 1.04 | NS |
| Rancidity | 0.26 | 0.10 [NS] | 0.16 [NS] | 0.23 [NS] | 0.52 | NS |
| Off-tastes | 0.33 | 0.20 [NS] | 0.27 [NS] | 0.31 [NS] | 0.61 | NS |
| Chewiness | 5.93 | −0.28 [NS] | 0.29 [NS] | −0.19 [NS] | 1.35 | NS |
| Softness | 6.03 | −0.51 [NS] | 0.23 [NS] | −0.49 [NS] | 1.21 | NS |
| Juiciness | 6.05 | −0.59 [NS] | −0.07 [NS] | −0.68 [NS] | 1,63 | NS |
| Pastiness | 0.32 | 0.14 [NS] | 0.08 [NS] | 0.14 [NS] | 0.39 | NS |

[a] The results are presented as differences (least squares means) between the control group and the tested formulation. CON = control with 110 mg/kg sodium nitrite and 0.2% phosphates; 50NI = 50% less sodium nitrite than in control; 50NA = 50% less sodium nitrite than in control with added 225 mg/kg sodium ascorbate; NP = no phosphates and 110 mg/kg sodium nitrite; RMSE = root mean square error of the model; F: formulation; Y90 = force decay coefficient. Significance: NS = $p > 0.10$; ‡ = $p < 0.10$; * = $p < 0.05$.

## 4. Conclusions

Nitrite reduction was associated with lower moisture loss due to greater microbial growth on the surface of the sausages and higher oxidation. The nitrite reduction tested did not affect microbiological safety. The addition of ascorbate to the nitrite-reduced formulation reduced oxidation and improved color stability after prolonged exposure to air. The absence of phosphates did not affect moisture loss but was associated with lower pH, greater oxidation, and differences in color. The observed effects on physicochemical properties did not result in differences that could be detected by the evaluators during sensory testing. It can be concluded that the formulations tested, particularly the combination of nitrite reduction and ascorbate addition or omission of phosphates, resulted in acceptable products. However, only certain amounts or combinations of additives were tested in the present study, and this should be followed up in future studies. In addition, further studies can also be conducted that include more in-depth microbiological testing (spoilage bacteria), chemical examination (protein oxidation), prolonged storage under various conditions and determination of shelf life.

**Author Contributions:** Conceptualization, M.Č.-P. and M.Š.; methodology, M.Č.-P. and M.Š.; validation, M.Č.-P. and M.Š.; investigation, M.Č.-P., M.Š. and M.O.; resources, M.Č.-P., M.Š. and M.O.; data curation, M.Č.-P., M.Š. and M.O.; writing—original draft preparation, M.Š. and M.O.; writing—review and editing, M.Č.-P., M.Š. and M.O.; visualization, M.Č.-P., M.Š. and M.O.; supervision, M.Š.; project administration, M.Č.-P. and M.Š.; funding acquisition, M.Č.-P. All authors have read and agreed to the published version of the manuscript.

**Funding:** This research was funded by Slovenian Research Agency, grant number P4-0133.

**Institutional Review Board Statement:** Not applicable.

**Informed Consent Statement:** Informed consent was obtained from all subjects involved in the study.

**Data Availability Statement:** Not applicable.

**Acknowledgments:** The authors would like to thank the family farm Ozmec for the help in conducting the experiment with sausage processing and to Maja Prevolnik Povše (Faculty of Agriculture and Life Sciences) and Urška Tomažin (Agricultural Institute of Slovenia) for their help with chemical analyses. The study was a part of the MSc thesis of Manja Ozmec.

**Conflicts of Interest:** The authors declare no conflict of interest.

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
