# Peer review of "Reduced Use of Nitrites and Phosphates in Dry-Fermented Sausages"

_processes, doi:10.3390/pr10050821_

Round 1

Reviewer 1 Report

Comments to the Author

This paper deals with the effects of reduced use of nitrites and phosphates on the quality of dry-fermented sausages. The subject is interesting. However, there are some problems as follows:

-The abstract needs rewriting since general information is not included in the abstract, numerical results should be included. Also, sensory properties and microbiological values are missing.

-Specific writing pertaining to the reported work is suggested.

-How the authors decide to use the mentioned concentrations of the nitrites and phosphates in the study? The methodology should be clearly explained in the abstract section, mold growth and oxidation are missing!

-At the end of the abstract section: add concise conclusions in one or two sentences.

- There is so much research in this field. The references are not new. Please refer to the more recent references. You should involve the motivation and the recent problems on the subject.

-I suggest dividing the text into several paragraphs with separated headings for each experiment.

- Are the ratios of nitrites and phosphates used in the formulation of sausages were by weight? If so, please indicate that.

- In the materials and methods section, authors should state the names of the company, city, and country for all materials and equipment.

-Line 91-98: Once defined, abbreviations should be systematically used. Here, CON, 50N, 50NA, NP, etc. Please check the whole manuscript for this kind of formatting mistake.

-Line 118: Please add the employed colour system: CIELAB ? if so, please define the main coordinates (in CIELAB: L*, a*, b*).

-Authors should include the brand and specifications of all equipment used in this study.

-Line 147: Please change “men” to male.

-Please provide literature references for each test; the formulation of the sample the section of methods.

-Please mind the word spacing! Check the whole manuscript, eg; L158, 254 etc.

-  There are so many researches in this field. The references are not new. Please refer to the more recent references. You should involve the motivation and the recent problems on the subject, and how could you differentiate between your work and the other published studies in the same field for eg;

Berardo, A., De Maere, H., Stavropoulou, D. A., Rysman, T., Leroy, F., & De Smet, S. (2016). Effect of sodium ascorbate and sodium nitrite on protein and lipid oxidation in dry fermented sausages. Meat Science, 121, 359-364.

-In addition, how is the current study different from the mentioned references [23,24,25]? authors should highlight the novelty of their work, clearly.

-Line 359: Please combine this sentence with the previous paragraph.

-The conclusions should reflect upon the aims - whether they were achieved or not. Please rewrite and acknowledge limitations and make recommendations for future work.

Author Response

Dear Reviewer #1, 

Thank you for your valuable time in commenting our manuscript. Answers to your comments/questions are listed below (in italics) right next to your text and the actual changes are visible in the corrected version of the manuscript using "track changes" option:   

This paper deals with the effects of reduced use of nitrites and phosphates on the quality of dry-fermented sausages. The subject is interesting. However, there are some problems as follows:

-The abstract needs rewriting since general information is not included in the abstract, numerical results should be included. Also, sensory properties and microbiological values are missing.Specific writing pertaining to the reported work is suggested.

Answer: The abstract was rewritten to include more specific results and numerical results were inserted when applicable. In the case of sensory properties there was no important differences and in the case of microbiological investigations the resulta were negative, so this was mentioned. Please see lines 9-29 of the revised text.   

-How the authors decide to use the mentioned concentrations of the nitrites and phosphates in the study?

Answer: As to the nitrite concentrations, concentrations (110 mg/kg) are based on the amount of Na-nitrite in the (pre-prepared and used) NaCl and Na-nitrite mixture (which is 0.5%). For dry-feremented sausages, the original recepie requests 2.2% of this salt mixture is added to the meat batter, resulting in concentration of 110 mg/kg Na-nitrite (i.e. 0.5% of 2.2%). As to the amount of phosphates used, we followed the instructions of the producer for such kind of product. This info was also added to the text (see lines  117-120 of the corrected text).

The methodology should be clearly explained in the abstract section, mold growth and oxidation are missing!

                Answer: Added to the abstract (please see lines 16 and 18 of the revised text).

-At the end of the abstract section: add concise conclusions in one or two sentences.

                Answer: Conclusions added as the text was rewritten. Please see lines 25-29 of the revised manuscript.

- There is so much research in this field. The references are not new. Please refer to the more recent references. You should involve the motivation and the recent problems on the subject.

                Answer: Several newer references were included along with the explanation of knowledge gaps. Please see lines 69-71, 80-82, 85-86, 97-100 and the reference list (i.e. references 18, 20, 31, 32) of the revised manuscript.

-I suggest dividing the text into several paragraphs with separated headings for each experiment.

                Answer: As already suggested by another Reviewer, the text (Material and Methods, Discussion) was divided and headings added. Please see lines 110, 138, 159, 170, 182, 209, 227, 235, 364, 395, 409, 426, 451 of the revised manuscript.

- Are the ratios of nitrites and phosphates used in the formulation of sausages were by weight? If so, please indicate that.

Answer: Yes. The info was added to the text (line 130)

- In the materials and methods section, authors should state the names of the company, city, and country for all materials and equipment.

                Answer: The text was checked and the infos added, please see lines 118, 127, 162, 185, 196 of the corrected text.

-Line 91-98: Once defined, abbreviations should be systematically used. Here, CON, 50N, 50NA, NP, etc. Please check the whole manuscript for this kind of formatting mistake.

                Answer: The text was checked and corrected throughout the text.  

-Line 118: Please add the employed colour system: CIELAB ? if so, please define the main coordinates (in CIELAB: L*, a*, b*).

                Answer: Yes it was CIELab, the info was added and parameters defined. Please see lines 160-163 of the corrected text.

-Authors should include the brand and specifications of all equipment used in this study.

                Answer: The text was checked and the infos added, please see lines 162,185,196 of the corrected text.

-Line 147: Please change “men” to male.

                Answer: Changed, please see line 211 of the corrected manuscript.

-Please provide literature references for each test; the formulation of the sample the section of methods.

                - Answer: Literature references for the used methods were added where applicable, the formulation of the samples was further explained, plese see lines 114-129, 163, 189, 210, 231-234  of the corrected manuscript. 

-Please mind the word spacing! Check the whole manuscript, eg; L158, 254 etc.

                Answer: The text was checked and corrected for word spacing and other language and style mistakes.

- There are so many researches in this field. The references are not new. Please refer to the more recent references. You should involve the motivation and the recent problems on the subject, and how could you differentiate between your work and the other published studies in the same field for eg; Berardo, A., De Maere, H., Stavropoulou, D. A., Rysman, T., Leroy, F., & De Smet, S. (2016). Effect of sodium ascorbate and sodium nitrite on protein and lipid oxidation in dry fermented sausages. Meat Science, 121, 359-364

Answer: We agree that there are already numerous studies on this field and that it is difficult to find some novelty with this kind of research. But, as also added to the manuscript, the present study provides an approach that is relatively broad (a more holistic aproach, covering many different aspects from the start of production to final quality-sensory traits), which many of these studies don't include. The study of Berardo et al., for instance, focuses more on oxidation, whereas studies like those of Hospital et al. ([23,24,25]) focus more on microbiological traits. These aspects, along with some newer references were included in the study, please see lines 69-71, 80-82, 85-86, 97-100 of the corrected manuscript.

-In addition, how is the current study different from the mentioned references [23,24,25]? authors should highlight the novelty of their work, clearly.

                Answer: Please see the answer to the previous comment.

-Line 359: Please combine this sentence with the previous paragraph.

Answer: Combined, please see lines 460 of the corrected manuscript.

-The conclusions should reflect upon the aims - whether they were achieved or not. Please rewrite and acknowledge limitations and make recommendations for future work.

                Answer: The conclusions were rewritten, limitations and future recommendations were added (lines 464-478 of the corrected manuscript).

Reviewer 2 Report

All comments are enclosed in the Manuscript.

In general, there is a need to:

  1. describe experimental setup in understandable manner,
  2. significance level add through the text in Results and Discussion sections,
  3. have a clear Results and Discussion without repeating what is already written in the Results.
  4. language editing.

Author Response

Dear Reviewer #2

Thank you for your valuable time in commenting our manuscript. Answers to your comments/questions are listed below (in italics) right next to your text and the actual changes are visible in the corrected version of the manuscript using "track changes" option:   

In general, there is a need to:

  1. describe experimental setup in understandable manner,

Answer: The experimental setup description was thoroughly rewritten, please see answers to specific commnets below.

  1. significance level add through the text in Results and Discussion sections,

Answer: Significance level was added, please see lines 271, 293, 296, 299, 300, 302, 316, 317, 320, 321, 322, 334, 336, 338, 341, 353 of the corrected text.

  1. have a clear Results and Discussion without repeating what is already written in the Results.

Answer: Repetitions in Discussion part were omited, please see lines 410-413, 428, 438-440 of the corrected text

  1. language editing.

Answer: Language was checked and corrected for mistakes, please see also answers to specific commnets below.

Reviewer's specific comments (as written directly in the text):

Line 9. Is this correct term?

Answer: Yes, referring to (according to the definition of the word) something that is »man-made« and not natural.

Line 101: The experimental design is not well described. How many sausages were produced per treatment?

Answer: There were 4 treatments (»preservation treatments«) each with 2 technical repetitions of 12-15 sausages, making it 24-30 sausages per treatment. The text was changed in order to better explain the experimental design. Please see lines 111-136 of the corrected manuscript.

Line 102: Preservation treatment?

                Answer:Please, see previous answer.

Line 113: This is not clear, first you say six sausages, then 12 per treatment.

                Answer: The dupalication in number is due to 2 technical repetitions per each of the treatments. To avoid ambiguity, the text was modified. Please see lines 111-136  of the corrected manuscript.

Line 119: On a production day? Before curing? Clarify.

                Answer:The measurement was performed on the finalised product (cured sausages). The explanation was added. Please see lines 163 of the corrected text.

Line 120: Not clear what samples? Sausages on a production day? If yes, were they stored whole or sliced?

                Answer: The samples ment here were cured sausages stored sliced. The explanation was added. Plesae see lines 165 of the corrected text.

Lines 169-170: Not clear from the text above what is treatment and what is technical repetition? Write above per treatmnet how many sausages was produced in total. And how many per batch as a technical replication of the treatment?

                Answer: This information was added. Please see lines 116 of the corrected manuscript.

Lines 171-173: It is not clear how MIXED model analysis were performend. What was random and what was fixed factor?

                Answer: This information was added, please see lines 245 of the corrected text.

Line 200: Is it neccessary to explain this abbreviation? Please throughout the whole manuscript pay attention to (), take the text from brackets into the sentence.

                Answer: Accoroding to the instructions for authors, saying: »When defined for the first time, the acronym/abbreviation/initialism should be added in parentheses after the written-out form« such abbreviations should be defined. As to the comment in regard to the (), this was corrected when possible.

Line 216: In general, p-values are missing in the text.

                Answer: p-values were added to the text, please see lines 271, 293, 296, 299, 300, 302, 316, 317, 320, 321, 322, 334, 336, 338, 341, 353 of the corrected text.

Line 232: Please rewrite this, as »-» is making confusion, specially in addition with the »+-», what should that mean.

                Answer: The »-» was changed to »=« to be in line with the rest of the table subheadings. The sign »+« was changed to »‡« to avoid ambiguity. Please see lines 310-311 and also subheadings of other tables in the corrected text.

Line 236: Please be consistent through the text, is it a treatment, formulations, technical replications…bit of confusion here.

                Answer: The text was checked and corrected for consistency.

Line 241: What should this mean?

                Answer: It was meant higher statistical significance. But to avod ambiguity, this part of the text was deleted, please see lines 321-322 of the corrected text.

Lines 253-259: Please rewrite.

                Answer: The comment is not specific enough, anyhow in accordance with the other comments of the reviewer (i.e. inserting p-values, excluding »()«), the text was rewritten. Please see lines 333-341 of the corrected text.

Lines 269-270: Be consistent! First of all, define each significance level on the text below Table. P lower than 0.1. is not significant, this can not see whay authors used this level of significance?

                Answer: Levels of significance were defined under each table. P-values below 0.1 were defined as tendencies and described/commented as such. Please see lines 351-354 of the corrected text.

Line 295: Remove.

                Answer: Removed, please see lines 381 of the corrected text.

Line 295: Please remove the brackets in the manuscript when not needed.

                Answer: The brackets were removed here and throughout the text. Please see the comment for line 200.

Line 298: What does this mean? Please rewrite the whole sentence.

                Answer: The sentence was rewritten, please see lines 384-385 of the corrected text.

Lines 299-300: It is not clear what do you want to say here. Rewrite.

                Answer: The text was rewritten, please see lines 386-387 of the corrected text.

Lines 307-309: Please rewrite, not clear what author wants to achieve here.

                Answer: The text was rewritten, please see lines 396-399 of the corrected text.

Line 309: This ref is about ADI for nitrite. Use the correct reference

Answer: The text was rewritten and the correct reference (now 24) cited. Please see lines 399-404 of the corrected text.

Line 310: Is this correct?

                Answer: The text was checked and rewritten, please see answer to the previous comment.

Lines 317-319: This was already said in line 217-218, please revise.

                Answer: This part was rewritten. In this case (i.e. results written seperately from discussion) not to mention shortly the results again – this enables much easier reading and understanding of the discussion. For the correction, please see lines 410-413 of the revised text.

Lines 322-323: Already written in line 219-220.

                Answer: The text was rewritten in order to eliminate repeating the results. Please see lines 417-419 of the corrected text.

Lines 329-340: Don't repeat results into discussion, please revise from line 329-340.

                Answer: The text was rewritten in order to eliminate repeating the results. Please see lines 428, 438-440 of the corrected text.

Reviewer 3 Report

The manuscript described the effects of reduced nitrites and phosphates additives on dry fermented sausages. Generally, some descriptions are not very clear. The introduction almost introduced the merit of adding nitrites and phosphate, so why we should reduce those additives (although we know the nitrites are toxic when used in high doses). The description of material and methods are very poor. The authors need major revision if it is considered in publishing in this journal. The specific comments are as follows:

Introduction: as aforementioned, the reason to reduce the doses of nitrites, nitrates, and phosphates in sausages was not well described. There was a contradictory statement in the introduction as well. For example, the authors stated the negative effects of chemical additives such as higher cost and especially lower efficacy in Line 74, then, what is the point to add ascorbate (as a replacement antioxidant) in your study? Because it is also regarded as a chemical additive. 

Lines 38-39: What do you mean by this sentence? Please rewrite it.

Lines 56-58: What do you mean by this sentence? Please rewrite it.

As there are several relevant research studies such as: "A study on the toxigenesis by Clostridium botulinum in nitrate and nitrite-reduced dry fermented sausages", "Surface treatment with condensed phosphates reduced efflorescence formation on dry fermented sausages with alginate casings" and so on, what the novelty of your research should be stated in the introduction as well.

Materials and Methods: the description of this part is rather weak and poor. Please be tidier and don't mix the method during the description. Each method should be placed and organized in one subtitle. For example, the authors described pH in Line 104, then it jumped to Line 133-135 again to describe the pH measurement. Please organize it into one section. It happens to other measurements as well. Please be more logical. The measurement is pH is not a common method for the pH measurement of sausage, what references did you refer to? 

The method of TBARS should be described in detail. 

The design of the experiments is not well described. Please provide the graph to better explain. Did you do the control (i.e. the samples with nitrite alone, or samples with phosphate alone)? In Lines 91-92, it seems the control group was mixed with sodium nitrite and phosphate. The volume added was not consistent with the results where CON = control with 110mg/kg sodium nitrite and 0.2% phosphates, I am not sure how the 110 mg/kg comes from. Also In line 97, why did you use 225mg/kg sodium ascorbate rather than other amounts? 

It is better to use other characters rather than 50N, as N is like the unit of force. 

Please combine the results and discussion for avoiding the repetition of the description.

Author Response

Dear Reviewer #3

Thank you for your valuable time in commenting our manuscript. Answers to your comments/questions are listed below (in italics) right next to your text and the actual changes are visible in the corrected version of the manuscript using "track changes" option:   

The manuscript described the effects of reduced nitrites and phosphates additives on dry fermented sausages. Generally, some descriptions are not very clear. The introduction almost introduced the merit of adding nitrites and phosphate, so why we should reduce those additives (although we know the nitrites are toxic when used in high doses).

                Answer: Along with the negative health aspects, also negative consumer attitude is the reason, which is mentioned in the introduction. Descriptions in the introduction were rewritten (also in according to the comments of the other two reviewers  – see lines 50-53, 66-71, 92-93, 97-100 of the revised text. 

The description of material and methods are very poor. The authors need major revision if it is considered in publishing in this journal.

                Answer: The Material and methods section was thoroughly revised and rewritten (also as to the comments of the other two reviewers). Please see lines 110-247 of the revised text.

The specific comments are as follows:

Introduction: as aforementioned, the reason to reduce the doses of nitrites, nitrates, and phosphates in sausages was not well described. There was a contradictory statement in the introduction as well. For example, the authors stated the negative effects of chemical additives such as higher cost and especially lower efficacy in Line 74, then, what is the point to add ascorbate (as a replacement antioxidant) in your study? Because it is also regarded as a chemical additive. 

                Answer: Ascorbate, although synthetically produced, is also present in foods as natural compound (vitamin C) and may therefore be (apart from not exerting toxic side effects) more acceptable to consumers. Please see lines 68-71 of the revised text.     

Lines 38-39: What do you mean by this sentence? Please rewrite it

Answer: It was ment that nitrate is converted (chamically reduced) to nitrite by notrate reductase. Now rewritten, please see lines 45-46 of the corrected text.

Lines 56-58: What do you mean by this sentence? Please rewrite it.

Answer: It was ment that ascorbate was not toxic, except for really high doses, which are practically impossible to reach. Now rewritten, please see lines 66-67 of the corrected text.

As there are several relevant research studies such as: "A study on the toxigenesis by Clostridium botulinum in nitrate and nitrite-reduced dry fermented sausages", "Surface treatment with condensed phosphates reduced efflorescence formation on dry fermented sausages with alginate casings" and so on, what the novelty of your research should be stated in the introduction as well.

                Answer: In line with the comment of the Reviewer 1, pointing out the similar issue, introduction was altered accordingly (please see lines 69-71, 80-82, 85-86, 97-100  of the revised manuscript) defining novelty and involving relevant newer references.

Materials and Methods: the description of this part is rather weak and poor. Please be tidier and don't mix the method during the description. Each method should be placed and organized in one subtitle. It happens to other measurements as well. Please be more logical.

 For example, the authors described pH in Line 104, then it jumped to Line 133-135 again to describe the pH measurement. Please organize it into one section.

                Answer: The description of pH measurement was joined together as well as other measurements (plese see lines 139-145). The Material and methods section was reorganised as suggested. Please see lines 110-247 of the corrected text.

The measurement is pH is not a common method for the pH measurement of sausage, what references did you refer to? 

                Answer: The reference was added, please see line 145.

The method of TBARS should be described in detail. 

                Answer: Detailed description of the method was added. Plese see lines 194-208 of the corrected text.

The design of the experiments is not well described. Please provide the graph to better explain.

                Answer: As already explained in the answers above and in the response to Reviewer #2, this section was rewritten and reorganizet in order to be more clear. An additional table (Table 1) was also intorduced in regard to the analyses involved. Please see lines 110-158 of the adapted manuscript.

Did you do the control (i.e. the samples with nitrite alone, or samples with phosphate alone)?

                Answer: The control group (CON) was prepared with nitrite and phosphate together, as this was the standard mixture used by the sausage producer. The fourth group - without phosphates (NP) was, however, prepared with nitrite alone (lines 118-119, 128-129)

In Lines 91-92, it seems the control group was mixed with sodium nitrite and phosphate. The volume added was not consistent with the results where CON = control with 110mg/kg sodium nitrite and 0.2% phosphates, I am not sure how the 110 mg/kg comes from.

                Answer:The amount of nitrite added was associated to nitrite salt (i.e. NaCl with 0.5% Na-nitrite) à in total 2.2% of nitrite salt was added to meat batter, resulting in 110 mg Na-nitrite/kg batter (0.5% of 2.2%). This info was also added to the text. Please see lines 117-119 of the corrected text.

Also In line 97, why did you use 225mg/kg sodium ascorbate rather than other amounts? 

                Answer: This was done according to the producer recommendations.

It is better to use other characters rather than 50N, as N is like the unit of force. 

                Answer: 50N was changed to 50NI throughout the entire manuscript.

Please combine the results and discussion for avoiding the repetition of the description.

                Answer: The issue of repeating the results was alredy raised by reviewer #2 and corrected accordingly. We presented separately the results and discussion section, as following the pre-defined word template of the journal, and would like to stick to that. For the corrections, please see lines  of the corrected manuscript. Please see lines 410-413, 428, 438-440 of the corrected text

Round 2

Reviewer 1 Report

The authors did a significant improvement which is satisfactory. I do not have any further comments and I am pleased to suggest it for publication in its current form in the Processes.

Author Response

The authors did a significant improvement which is satisfactory. I do not have any further comments and I am pleased to suggest it for publication in its current form in the Processes.

Answer: we would like to thank the reviewer to the time and effort dedicated to our article. The language was re-checked with Instatext.

Reviewer 3 Report

The manuscript improved to some extent but there are still some points that did not address my comments.

The authors stated that "Along with the negative health aspects, also negative consumer attitude is the reason, which is mentioned in the introduction.", but in Line 50-53, I don't see the negative health aspects.

For the measurement of pH, the authors made efforts to improve the explanation, unfortunately, it becomes more confusing. I don't know why there should be one sausage per technical replicate, what do you mean by "technical replicate"? In Line 142-432, I don't understand why you would like to use one sausage per technical replicate during processing and for 6 sausages per technical replicate at the end. Perhaps, the flowchart will be easy to understand.

For results, it is highly recommended to combine results and discussion, to avoid repetition and easy to understand. For the "changes during sausage processing, it is too general and why didn't you put mold coverage and microbiological attribute together? Please add x-axis and y-axis titles for all the figures.  For the statistical analysis, it is very common to use P<0.05, or P<0.01, normally, P>0.05 was regarded as not significant while the authors used P<0.1 for evaluating the data in Table 5. Please state the reason. Please uniform the standard of significant level in all tables. For example, Table 4 and Tables 5, 6 are different.

Author Response

We would like to thank the reviewer to the time and effort dedicated to our article. The answers to the comments are provided below and the changes to the document are visible in the text using »track changes option of Word software.

The manuscript improved to some extent but there are still some points that did not address my comments. The authors stated that "Along with the negative health aspects, also negative consumer attitude is the reason, which is mentioned in the introduction.", but in Line 50-53, I don't see the negative health aspects.

Answer: Sorry, this was our mistake- negative health aspects are mentioned in the lines 47-51(before lines 56-60).

For the measurement of pH, the authors made efforts to improve the explanation, unfortunately, it becomes more confusing. I don't know why there should be one sausage per technical replicate, what do you mean by "technical replicate"? In Line 142-432, I don't understand why you would like to use one sausage per technical replicate during processing and for 6 sausages per technical replicate at the end. Perhaps, the flowchart will be easy to understand.

Answer: The term»technical replicate« refers to the batch (we did 2 batches of the same formulation). With regard to the monitoring of pH through processing, one sausage was used (sacrificied). For statistical assessment of the final pH, 6 sausages per batch wre used, as in the case of other physico-chemical properties. A flowchart was added to the manuscript (Figure 1) to more clearly explain the experimental design and the number of samples used, and Material & Methods corrected accordingly (see lines 98-222 of the corrected text).

For results, it is highly recommended to combine results and discussion, to avoid repetition and easy to understand. For the "changes during sausage processing, it is too general and why didn't you put mold coverage and microbiological attribute together?

Answer: Results and Discussion are now merged, as suggested. Mold coverage and microbiological tests are also joined in the same paragraph, as suggested.

Please add x-axis and y-axis titles for all the figures.

Answer: The titles for both axes were added to the figures (please see Figures 2 and 3 in the corrected text).

For the statistical analysis, it is very common to use P<0.05, or P<0.01, normally, P>0.05 was regarded as not significant while the authors used P<0.1 for evaluating the data in Table 5. Please state the reason. Please uniform the standard of significant level in all tables. For example, Table 4 and Tables 5, 6 are different.

Answer: We agree with the reviewer, that generally P<0.05 denotes a significant effect. Thus when referring to P<0.1 we always use »tendency« (towards statistical significance). However, we found one inconsistency (line 369) - now corrected). Tables are also uniformed in that regard.

Round 3

Reviewer 3 Report

The authors addressed most of my comments. Just a small revision before it can be accepted for publication. The number before "Conclusion" should be 4.